# Changes in Smoking Behaviour and Home-Smoking Rules during the Initial COVID-19 Lockdown Period in Israel

**DOI:** 10.3390/ijerph18041931

**Published:** 2021-02-17

**Authors:** Yael Bar-Zeev, Michal Shauly-Aharonov, Hannah Lee, Yehuda Neumark

**Affiliations:** Braun School of Public Health and Community Medicine, Hebrew University-Hadassah Medical Centre, Jerusalem 9112102, Israel; michal.shauly@mail.huji.ac.il (M.S.); lee.hannah15@gmail.com (H.L.); yehudan@ekmd.huji.ac.il (Y.N.)

**Keywords:** smoking, COVID-19, smoking cessation, home-smoking-rules

## Abstract

The COVID-19 pandemic has caused devastating impacts globally. To mitigate virus spread, Israel imposed severe restrictions during March–April 2020. An online cross-sectional survey was conducted in April 2020 among current and ex-smokers to explore changes in smoking behaviour and home-smoking rules during this period. Bivariate analysis and multivariate logistic regression examined associations between sociodemographic characteristics and perceived risk of infection and quitting smoking during the initial COVID-19 period. Current smoking was reported by 437 (66.2%) of the 660 participants, 46 (7%) quit during the initial restriction period, and 177 (26.8%) were ex-smokers. Nearly half (44.4%) of current smokers intensified their smoking, and 16% attempted to quit. Quitting during the COVID-19 period was significantly associated with higher education (adjusted odds ratio (aOR): 1.97, 95% CI: 1.0–3.8), not living with a smoker (aOR: 2.18, 95% CI: 1.0–4.4), and having an underlying chronic condition that increases risk for COVID-19 complications (aOR: 2.32, 95% CI: 1.1–4.6). Both an increase in smoking behaviour and in attempts to quit smoking during the initial COVID-19 pandemic were evident in this sample of adult Israeli smokers. Governments need to use this opportunity to encourage smokers to attempt quitting and create smoke-free homes, especially during lockdown conditions, while providing mental and social support to all smokers.

## 1. Introduction

The SARS-CoV-2 coronavirus pandemic has caused devastating health, social, and economic impacts globally [1]. Many countries placed extreme restrictions and lockdowns on residents in an attempt to contain the virus’ spread [2]. In Israel, restrictions on the entrance into the country went into effect on 22 January 2020. More extensive restrictions on residents began on 4 March, escalating to an almost complete lockdown imposed on 19 March. Relaxation of restrictions began a month later on 19 April 2020, but extensive relaxations began only on 3 May [2].

Older age, male gender, and underlying chronic conditions, such as cardiovascular disease, chronic obstructive pulmonary disease, and diabetes, are all purported risk factors for severe disease and mortality from the SARS-CoV-2 virus [3,4,5,6]. Tobacco use is an independent risk factor for most of the underlying chronic conditions implicated as possible risk factors for severe SARS-CoV-2 infection [7]. In addition, tobacco use is known to suppress the immune system and increase the risk for various respiratory infections [8,9]. Tobacco use has also been suggested as a possible risk factor for severe disease in previous coronavirus outbreaks, such as the Middle East Respiratory Syndrome [10,11,12].

Angiotensin-Converting Enzyme-2 (ACE2) receptors in the alveolar lung tissue are the primary route of entry of the SARS-CoV-2 virus into the human body [13,14]. Analysis of ACE2 in the lungs found that ACE2 expression is upregulated in heavy smokers [13,14,15]. However, other studies have suggested that nicotine downregulates the ACE2 receptor [16]. Currently, there is uncertainty regarding the association between smoking and SARS-CoV-2 infection and severity [17]. Several studies have reported that current smokers have lower than expected risk for SARS-CoV-2 infection, [18,19,20,21,22] while other studies have suggested that hospitalized smokers with SARS-CoV-2 infection have an increased risk for a severe infection [3,23].

The mental health impacts of the pandemic, economic implications of the restrictions, social isolation and confinement to home, and perceptions of the health risks from SARS-CoV-2 infection may influence people to change their health behaviours, [24,25,26] including their smoking behaviour, either to increase their smoking, to reduce or quit smoking, or to change their home-smoking rules [27].

The aims of this study were to explore (1) changes in smoking behaviour during the initial COVID-19 period and the restrictions period in Israel and the factors associated with quitting smoking during these periods, and (2) changes in home-smoking rules during the initial COVID-19 period and the restrictions period in Israel.

## 2. Materials and Methods

Design: An online cross-sectional survey was conducted during the initial COVID-19 restrictions period in Israel (6–28 April 2020). Eligible participants were Israeli smokers—current or former, Hebrew speakers, and 18 years of age or older. The questionnaire was distributed through social media channels—promoted via Facebook and Instagram ads and shared without promotion on Twitter and WhatsApp. No incentive was offered for participation or completion of the survey.

Main study outcomes: (1) quitting during the initial COVID-19 restrictions period in Israel; (2) changes in home smoking rules during the initial COVID-19 restrictions period in Israel; and (3) changes in smoking behavior among current smokers.

Survey instrument (Appendix A): Participants were asked about their sociodemographic characteristics, current Sars-CoV-2 exposure and infection status, perception of risks for potential complications from Sars-CoV-2 infection, perceived stress levels, smoking status, changes in home-smoking rules, and smoking patterns (among current smokers only).

Sociodemographic characteristics included age, sex, education level (<12 years of school, 12 years, bachelor’s degree, or master’s degree or higher; re-categorised for the regression analysis as ≤12 years of schooling and bachelor degree and higher), religion (Jewish, Christian, Muslim, or other; re-categorised to Jewish and other), marital status (single, married, living with a partner, widowed, or divorced; re-categorised to married or living with a partner and single/widowed/divorced), employment status prior to the COVID-19 restrictions (full-time job, part-time permanent, part-time casual, self-employed, unemployed, or retired), employment status during COVID-19 restrictions (no change, reduction in working hours, unpaid leave of absence, loss of employment, or reduced income from business), any smoker living at home, number of children living at home and age of youngest child (recategorized to ≤6 years old and >6 years old), anyone at high risk for Sars-CoV-2 infection complications living at home (defined as old age and/or with any chronic disease), and outdoor home space (garden, balcony only, or no garden or balcony).

Sars-CoV-2 exposure or infection status was categorized as: not exposed to a confirmed case, exposed and currently in quarantine, past exposure and quarantine, currently ill with the virus, and past illness with the virus.

Perception of risk was measured based on four questions: two questions assessed the general perception of a smoker’s risk of infection with the SARS-CoV-2 virus, and if infected, the risk to develop severe illness (for both—smoker’s risk is higher, the same, or lower compared to non-smoker’s risk), and two questions pertained to perception of participant’s own personal risk of infection with the SARS-CoV-2 virus, and if infected, to develop severe illness (both using a Likert scale from 1–10, where one is no risk at all and 10 is very high risk). Underlying chronic illness (as a measure of possible personal risk for severe illness) was ascertained with a dichotomous (yes/no) question asking whether the participant had any chronic illnesses, including cardiovascular disease, chronic obstructive pulmonary disease such as bronchitis or emphysema, asthma, hypertension, diabetes, or cancer. Perceived mental stress was measured with two questions: (1) “Before the COVID-19 period, to what extent did you feel you were under mental stress?” (very low, low, medium, high, very high), and (2) “Since the COVID-19 period, how much do you feel that your mental stress level has changed?” (increased considerably, increased slightly, no change, reduced slightly, reduced considerably).

Smoking status was measured using two questions: (1) “Do you currently smoke?” (yes, every day; yes, sometimes; no, I used to smoke and I quit; no, I never smoked); and (2) “If you used to smoke and quit, when did you quit?” (since the start of the COVID-19 period; between 3–12 months ago—before the start of the COVID-19 period but less than a year ago; more than a year ago). From these two questions, participants were categorized to current smokers (daily and non-daily combined), quit during the COVID-19 period, and ex-smoker (quit before the COVID-19 period). In addition, participants were asked whether they used any other tobacco or smoking products (electronic cigarette, heated tobacco products, hookah or water pipe, and other)—for each option, yes/no.

Smoking patterns were assessed among current smokers only. Participants were asked about the average number of cigarettes smoked daily before the COVID-19 period; time to first cigarette in the morning pre-COVID-19 (within 5 min of waking, 6–30 min, 31–60 min, or over one hour); average daily number of cigarettes during the COVID-19 period; changes in smoking patterns during the COVID-19 period (I smoke more/the same/less than usual); motivation and self-efficacy to quit smoking prior to the COVID-19 period (10-point Likert scale from low (1) to high (10) motivation and self-efficacy); changes in motivation and self-efficacy to quit during the COVID-19 period as well as changes in frequency and strength of urges to smoke (increased considerably, increased slightly, no change, reduced slightly, or reduced considerably); attempted to quit smoking since the beginning of the COVID-19 period (yes/no); longest attempt (<24 h, 1–3 days, 4–7 days, 8–14 days, or >14 days); use of any support in the attempt to quit (respondents could report multiple forms of support)—for each option yes/no (Quitline, family physician, prescription medication, nonprescription medication, other, no support). As a measure of nicotine addiction level, the Heaviness of Smoking Index [28] (range 0–6) was computed based on the average number of cigarettes per day and time to first cigarette.

Home-smoking rules were categorized using two questions, one regarding the rules prior to the COVID-19 restrictions and one regarding the rules during the COVID-19 restrictions. For both questions, options included: smoking allowed in the entire house, only in certain rooms, in one room only, on the balcony only, or smoking not allowed anywhere in the house including the balcony. Answers to these two questions were used to create a composite scale: rules changed to reflect less exposure to smoking for other family members (e.g., changing from smoking being allowed in certain rooms to only being allowed in one room), rules did not change, or rules changed to reflect more exposure (e.g., from smoking not allowed anywhere in the house to smoking being allowed in one room).

Statistical Analysis: Descriptive analysis was conducted using frequencies (%) for categorial variables and mean (SD) for continuous variables. Bivariate analysis was performed using Pearson’s χ2 test for categorical measures (with post hoc Bonferroni correction for multiple comparisons), *t*-test for continuous measures, and Kruskal–Wallis for ordinal measures to examine the associations between sociodemographic characteristics and perceptions of risk of Sars-CoV-2 infection and quitting smoking during the COVID-19 initial restrictions period. Multivariate logistic regression models included as covariates all factors that were significant in the bivariate analysis, adjusted for gender and age to account for possible confounding. A *p*-value ≤ 0.05 was considered statistically significant. Analyses were performed using SPSS v25 (IBM, Armonk, NY, USA).

Ethics: The study was approved by the Ethical Committee for Scientific Research on Human Subjects at the Hebrew University-Hadassah Faculty of Medicine (approval #05042020).

## 3. Results

Overall, 687 people responded, and 685 provided consent to participating in our survey. Excluded from the study were 24 respondents who indicated that they were never smokers and one who was underage.

Of the 660 participants included in the below analyses, 437 (66.2%) were current smokers, 46 (7%) quit during the COVID-19 restrictions, and 177 (26.8%) were ex-smokers who quit prior to the COVID-19 restrictions. The use of other forms of tobacco or smoking products was reported by 19% of current smokers (*n* = 83), 19.8% of ex-smokers (*n* = 35), and 8.7% (*n* = 4) of those who quit during the COVID-19 restrictions. The most common form of other product used was narghile (water pipe; *n* = 41, 6.2% of the total sample), followed by cannabis (nonmedicinal; *n* = 27, 4.1%), electronic cigarette (*n* = 24, 3.6%), and heated tobacco products (*n* = 18, 2.7%).

Sociodemographic characteristics: Table 1 presents the sociodemographic characteristics of the entire sample by smoking status. Compared to current smokers, those who quit during COVID-19 restrictions were more likely to be highly educated (43.5%; *n* = 20 and 26%; *n* = 112, respectively, *p* = 0.015) and less likely to have other smokers living in the house (28.3%; *n* = 13 and 47.3%; *n* = 204, respectively, *p* = 0.019). No other sociodemographic differences were noted between current smokers and those who quit during COVID-19 restrictions. Compared to current smokers and to those who quit during the initial COVID-19 restrictions period, ex-smokers (who quit prior to the COVID-19 restrictions) tended to be older, male, living with a partner, with their employment status unchanged during the initial COVID-19 restrictions period (Table 1).

Sars-CoV-2 exposure or infection status: Few respondents reported being exposed or infected with the Sars-CoV-2 virus (4.6%, *n* = 30), of whom 26 had been exposed to a confirmed Sars-CoV-2 case and went into quarantine, and 4 participants were currently or in the past infected with the virus.

Perception of risk, chronic illness, and stress levels: Just under half of the total sample (48%) perceived that smokers are at higher risk of Sars-CoV-2 infection, and 81.4% perceived smokers’ risk of a severe disease to be higher than non-smokers (Table 2). A higher proportion of those who quit during COVID-19 restrictions perceived smokers’ risk for severe disease to be higher, compared to current smokers—91.3% and 77.2%, respectively, *p* = 0.024. Additionally, the prevalence of a chronic disease was significantly higher among those who quit during the COVID-19 restrictions (37%, *n* = 17) than among current smokers (21.4%, *n* = 93) (*p* = 0.025). There was no difference between those who quit during the COVID-19 restrictions and current smokers in their perception of personal risk of infection nor personal risk of a severe disease (Table 2).

No differences in perceived stress levels prior to COVID-19 restrictions were noted between current smokers, those who quit during COVID-19 restrictions, and ex-smokers. As seen in Table 2, a higher proportion of those who quit smoking during COVID-19 restrictions reported a significant reduction in their perceived stress levels (8.7%), compared to current smokers (2.1%) or ex-smokers (2.8%), *p* = 0.002. At the same time, a lower proportion of ex-smokers reported that their stress levels increased considerably during COVID-19 restrictions (15.3%), compared to current smokers (29.6%) or those who quit during COVID-19 restrictions (23.9%), *p* = 0.002.

Home-smoking rules: Overall, for most participants (80.5%) no change occurred in their home-smoking rules during the COVID-19 restrictions, while for 12.9% the rules changed to reflect less exposure to other household members, and for 6.6% the rules changed to reflect more exposure to other household members. After COVID-19 restrictions went into place, three-quarters of those who quit smoking reported having a complete smoking ban in their homes (73.3%, *n* = 33) compared with 13.3% (*n* = 58) among current smokers (*p* < 0.001).

Smoking patterns among current smokers: As seen in Table 3, 44.4% of current smokers reported smoking more during the COVID-19 restrictions, and on average, increasing their cigarettes per day by 2.3 (SD 7.1), with no difference between those who did or did not attempt to quit. Over half of current smokers also reported an increase in smoking frequency (56.3%) and strength of urges to smoke (54.4%). Among those who attempted to quit (*n* = 70), 39.1% reported the attempt lasted less than 24 h, 37.7% up to a week, and 23.3% for more than a week. Among those who made quit attempts, 11 (15.7%) reported using some form of support (intensive behavioural support such as Quitline *n* = 5; prescription medication *n* = 5; nonprescription medication *n* = 3; family physician support *n* = 4).

Factors associated with quitting during COVID-19 restrictions: Quitting during the COVID-19 restrictions was associated with having earned a bachelor’s degree or higher (adjusted odds ratio (aOR): 1.97; 95% CI 1.0, 3.8; *p* = 0.048), not living with a smoker in the home (aOR: 2.18; 95% CI 1.0, 4.4; *p* = 0.032), and having a chronic disease (aOR: 2.32; 95% CI 1.1, 4.6; *p* = 0.017) (Table 4).

## 4. Discussion

In this online sample of Israeli smokers, both an increase in smoking behaviour and an increase in motivation to quit, quit attempts, and smoking cessation during the COVID-19 restrictions period were reported. Among current smokers, 44% reported an increase in the number of cigarettes smoked, and over 50% reported an increase in the frequency and in the strength of urges to smoke. On the other hand, 45% reported an increase in their motivation to quit. One-quarter of respondents who smoked at the beginning of the COVID-19 period in Israel reported quit attempts, both successful and unsuccessful. Having a higher educational degree, not living with another smoker in the home, and having an underlying chronic illness were all associated with quitting during the initial COVID-19 restrictions period in Israel. Surprisingly, both pre-COVID-19 stress levels and perceived changes in stress levels during the COVID-19 restrictions period were similar among current smokers and those who quit.

Survey questions regarding quit attempts were not specific to quitting due to COVID-19, and participants may have quit or attempted to quit for other reasons. Nonetheless, the baseline level of quit attempts per year among smokers in Israel is approximately 20% [29], suggesting that at least some of the increase was due to COVID-19.

Similar to our results, Klemperer and colleagues (2020) [27] found that smokers had varying responses to the COVID-19 pandemic. In their survey of smokers in the US, over 20% reported a quit attempt and over 30% reported an increase in their tobacco use. Our findings are also similar to an online survey in Poland in which 45% of smokers reported smoking more during the last two weeks of April [30]. Online cross-sectional surveys from Italy and Spain during the initial lockdown phase found that ~3% reported quitting smoking [26,31]. On the other hand, a Google Trends analysis did not show an increase in interest in smoking cessation during late February and March 2020 [32]. Similar to findings from the present study, a survey in Australia conducted during early to mid April found that respondents who reported a negative change in their smoking behaviour also reported higher levels of depression, anxiety, and stress [33].

The uncertainty regarding the role of smoking and nicotine on the risk for infection and COVID-19 severity might also impact smoking behaviors [34,35]. Several meta-analyses have suggested that former smokers are at increased risk for a severe disease [17,36,37], while other studies have shown that the proportion of smokers among infected patients was lower than expected [17,38]. This has raised the hypothesis that nicotine might play an immune-modulator protective role in the pathogenesis of the disease [39,40]. Claims based on this hypothesis were made on popular media outlets, which might impact smoking behavior [34,35]. Future studies assessing changes in smoking behavior need to also assess exposure to these claims and the impact of this exposure on the changes in smoking behavior.

The study sample was a convenience sample recruited through social media. In Israel 85% of all adults report using Facebook, 52% report using Instagram, and 15% report using Twitter [41]. The sample, therefore, is likely to not be fully representative of all smokers and ex-smokers in Israel [42]. More specifically, the sample was primarily restricted to Jewish participants, and females were over-represented (in comparison to their proportion among Israeli smokers). The ultraorthodox Jewish population in Israel (11% of the population) [43], who suffered several outbreaks of SARS-CoV-2 and do not use social media [44], is not represented in this survey. Information about smoking behaviour characteristics was only collected from current smokers, making it impossible to explore possible associations between smoking characteristics and quitting during the COVID-19 initial restrictions period in Israel. Certain smoking characteristics such as a lower nicotine dependence level, higher perceived self-efficacy to quit, and higher baseline motivation level might be associated with a greater likelihood of successful quitting during the initial COVID-19 restrictions period. Some evidence for this can be seen from findings presented in Table 3 comparing current smokers who attempted to quit and those that did not attempt to quit. Data regarding the proportion of former smokers who relapsed during the initial COVID-19 restrictions period were not collected. It is therefore not possible to explore which factors might impact relapse. In addition, the survey did not include a question on the number of years the respondent had been smoking, so we are unable to assess pack-years. This study focused on those smoking combustible tobacco products, with only a small proportion reporting also using other noncombustible products, such as electronic cigarettes. Electronic cigarette use in Israel is currently low at 1.8% [45]. Despite these limitations, our findings are comparable to another Israeli online survey conducted in late March 2020 among 297 Jewish smokers, in which 36% of smokers reported an increase in their smoking, and 3% reported quitting smoking [46]. The strengths of this study are the relatively large sample size and the early data collection during the COVID-19 restrictions, thus minimizing recall bias.

Findings from this study and similar studies in other countries suggest that two opposing trends were occurring among smokers during the COVID-19 initial restrictions period—with some people experiencing an increase in motivation to quit, and quit attempts, while others intensified their smoking behaviour, which might be related to an increase in stress levels. COVID-19 policies need to address both these subpopulations of smokers. On the one hand, seizing this opportunity to proactively offer more smoking cessation support to those seeking it, such as moving current face-to-face smoking cessation options to online platforms and encouraging use of Quitlines. In Israel, the most common form of smoking cessation support provided by Health Maintenance Organizations (HMO) is free group behavioural counselling, coupled with subsidized pharmacotherapy. However, during the COVID-19 restrictions, all group counselling was suspended; those who had commenced group therapy prior to the restrictions continued with personal one-on-one phone counselling, but no new online options were available for those seeking to quit during the initial restrictions period. A government sponsored national Quitline was launched in Israel in January 2020, but no media campaigns were conducted to raise awareness about the Quitline, and during the COVID-19 restrictions, the number of counsellors available at the Quitline were actually reduced to comply with social distancing requirements. On the other hand, governments and HMOs should also address the increase in smoking behaviours, especially the increase in smoking within the house, which exposes other household members to second-hand smoke. This could be done by providing online social and mental support options to those confined to their homes—initiating media campaigns to raise awareness about second-hand smoke risks and to create smoke-free homes, and also providing options for support not only to those who are interested in quitting but offering help to all smokers regardless of their motivation to quit. Innovative strategies should be explored, such as an HMO proactively contacting all smokers through text messaging to offer support and mailing out free nicotine replacement therapy options to help cope with urges to smoke. As countries continue struggling to cope with the SARS-CoV-2 pandemic, including relaxation of certain restrictions and imposing others (such as requiring use of facial masks), more longitudinal representative samples are needed to explore the effects of these regulations, and of the ongoing pandemic, on smoking behaviours.

## 5. Conclusions

During the initial phase of COVID-19 restrictions, both an intensification of smoking behaviours and an increase in attempts to quit smoking were evident in this sample of adult Israeli smokers. By raising awareness and providing online and remote cessation support options, governments and health organizations need to use this opportunity to encourage more smokers to attempt quitting and create smoke-free homes. In addition, there is a need to explore options for supporting those not interested in quitting, who may be struggling with their addiction due to the pandemic.

## Figures and Tables

**Table 1 ijerph-18-01931-t001:** Sociodemographic characteristics of the total sample (*N* = 660) and by smoking status, Israel, 2020.

	Total (*n*, %) *N* = 660	Smoking Status			
		Current Smokers (*n* = 437, 66.2%)	Quit during COVID-19 Restrictions (*n* = 46, 7%)	Quit before COVID-19 Restrictions (*n* = 177, 26.8%)	*p*-Value
Age mean years, (SD) (missing *n* = 22)	40.2 (14.55)	38.6 (14.57)	38.3 (12.84)	44.4 (14.11)	<0.001
Sex (missing *n* = 2)					
Female	397 (60.3%)	274 (62.8%)	32 (69.6%)	91 (51.7%)	0.016
Education (missing *n* = 6)					
<12 years	54 (8.3%)	41 (9.5%)	2 (4.3%)	11 (6.2%)	<0.001
12 years	384 (58.7%)	278 (64.5%)	24 (52.2%)	82 (46.3%)	
Bachelor’s degree	152 (23.3%)	85 (19.7%)	15 (32.6%)	52 (29.4%)	
Master’s degree or higher	64 (9.8%)	27 (6.3%)	5 (10.9%)	32 (18.1%)	
Religion (missing *n* = 9)					
Jewish	615 (94.5%)	406 (94.2%)	43 (93.5%)	166 (95.4%)	0.804 ^
Muslim	7 (1.1%)	3 (0.7%)	0 (0%)	4 (2.3%)	
Christian	15 (2.3%)	12 (2.8%)	1 (2.2%)	2 (1.1%)	
Other *	14 (2.2%)	10 (2.3%)	2 (4.3%)	2 (1.1%)	
Marital Status (missing *n* = 3)					
Married/Living with a partner	336 (51.1%)	203 (46.8%)	22 (47.8%)	111 (62.7%)	0.002
Single/Divorced/Widowed	321(48.9%)	231 (53.2%)	24 (52.2%)	66 (37.3%)	
Outdoor home space (missing *n* = 10)					
Garden	275 (42.3%)	163 (37.9%)	21 (45.7%)	91 (52.3%)	0.029
Balcony	235 (36.2%)	166 (38.6%)	16 (34.8%)	53 (30.5%)	
No balcony or garden	140 (21.5%)	101 (23.5%)	9 (19.6%)	30 (17.2%)	
Employment status prior to COVID-19 restrictions (missing *n* = 6)					
Full-time job	310 (47.4%)	207 (48%)	24 (52.2%)	79 (44.6%)	0.086
Part-time permanent	101 (15.4%)	71 (16.5%)	6 (13%)	24 (13.6%)	
Part-time casual	40 (6.1%)	33 (7.7%)	2 (4.3%)	5 (2.8%)	
Self-employed	64 (9.8%)	34 (7.9%)	5 (10.9%)	25 (14.1%)	
Not working/unemployed	96 (14.7%)	64 (14.8%)	6 (13%)	26 (14.7%)	
Retired	43 (6.6%)	22 (5.1%)	3 (6.5%)	18 (10.2%)	
Employment status change during COVID-19 restrictions (missing *n* = 17)					
No change	350 (54.4%)	219 (51.2%)	20 (45.5%)	111 (64.9%)	0.004
Reduced income (total):	293 (45.6%)	209 (48.8%)	24 (54.5%)	60 (35.1%)	
Reduced hours	58 (9%)	37 (8.6%)	7 (15.9%)	14 (8.2%)	0.001
Unpaid leave	164 (25.5%)	125 (29.2%)	11 (25%)	28 (16.4%)	
Loss of employment	23 (3.6%)	21 (4.9%)	1 (2.3%)	1 (0.6%)	
Self-employment income significantly reduced	48 (7.5%)	26 (6.1%)	5 (11.4%)	17 (9.9%)	
At least one child under 18 years old living at home (missing *n* = 46)	311 (50.7%)	215 (52.4%)	24 (54.5%)	72 (45%)	0.242
Age of youngest child living at home (among those with children under 18, *n* = 303, missing *n* = 8)					
<6 years	117 (38.6%)	74 (35.2%)	11 (47.8%)	32 (45.7%)	0.19
≥6 years	186 (61.4%)	136 (64.8%)	12 (52.2%)	38 (54.3%)	
Another smoker living at home (missing *n* = 7)	278 (42.6%)	204 (47.3%)	13 (28.3%)	61 (34.7%)	0.002
High risk individual for Sars-CoV-2 severe infection living at home (missing *n* = 6)	208 (31.8%)	138 (31.9%)	17 (37%)	53 (30.3%)	0.687

* Other religions include—No religion/Atheist/Buddhist. ^ Comparison for religion is between Jewish and all others combined.

**Table 2 ijerph-18-01931-t002:** COVID-19 risk perceptions by smoking status, chronic disease, and stress levels (*N* = 660), Israel, 2020.

	Total (*n*, %) *N* = 660	Current Smokers (*n* = 437, 66.2%)	Quit during COVID-19 Restrictions (*n* = 46, 7%)	Quit before COVID-19 Restrictions (*n* = 177, 28.6%)	*p*-Value
Perception of smokers’ risk for Sars-CoV-2 infection (missing *n* = 1)					
Higher risk	316 (48%)	205 (47%)	25 (54.3%)	86 (48.6%)	0.627
Same or lower risk	343 (52%)	231 (53%)	21 (45.7%)	91 (51.4%)	
Perception of smokers’ risk for severe Sars-CoV-2 infection (missing *n* = 3)					
Higher risk	535 (81.4%)	335 (77.2%)	42 (91.3%)	158 (89.3%)	<0.001
Same or lower risk	122 (18.6%)	99 (22.8%)	4 (8.7%)	19 (10.7%)	
Perception of personal risk for Sars-CoV-2 infection Mean (scale 1–5), (SD) (missing *n* = 5)	4.67 (2.19)	4.66 (2.19)	4.78 (2.14)	4.66 (2.2)	0.826
Perception of personal risk for severe Sars-CoV-2 infection Mean (scale 1–5), (SD) (missing *n* = 9)	4.88 (2.47)	5.1 (2.47)	5.09 (2.55)	4.26 (2.38)	0.001
Underlying chronic illness (missing *n* = 3)	157 (23.9%)	93 (21.4%)	17 (37%)	47 (26.7%)	0.037
Perceived stress level prior to COVID-19 restrictions (missing *n* = 3)					
Very low	124 (18.9%)	77 (17.7%)	11 (23.9%)	36 (20.5%)	0.56
Low	160 (24.4%)	104 (23.9%)	11 (23.9%)	45 (25.6%)	
Medium	192 (29.2%)	122 (28%)	13 (28.3%)	57 (32.4%)	
High	119 (18.1%)	86 (19.8%)	6 (13%)	27 (15.3%)	
Very high	62 (9.4%)	46 (10.6%)	5 (10.9%)	11 (6.3%)	
Perceived change in stress level during COVID-19 restrictions (missing *n* = 4)					
Increased considerably	166 (25.3%)	128 (29.6%)	11 (23.9%)	27 (15.3%)	0.002
Increased slightly	271 (41.3%)	175 (40.4%)	17 (37%)	79 (44.6%)	
Did not change	168 (25.6%)	99 (22.9%)	10 (21.7%)	59 (33.3%)	
Decreased considerably	33 (5%)	22 (5.1%)	4 (8.7%)	7 (4%)	
Decreased slightly	18 (2.7%)	9 (2.1%)	4 (8.7%)	5 (2.8%)	

**Table 3 ijerph-18-01931-t003:** Smoking patterns among current smokers (*N* = 437), Israel, 2020.

	Total (*n*, %) *N* = 437	Quit Attempt during COVID-19 Restrictions Period		
		Did Not Attempt to Quit Smoking (*n* = 362)	Attempted to Quit Smoking (*n* = 70)	*p*-Value
Regular smoker (≥1 cigarette/day)	400 (91.5%)	343 (94.8%)	52 (74.3%)	<0.001
Number of cigarettes/day before COVID-19 restrictions Mean (SD) (missing *n* = 57)	15.6 (9.9)	15.8 (10.2)	14.27 (7.7)	0.341
Time to first cigarette in the morning (missing *n* = 14)				
≤5 min	84 (19.9%)	71 (20.3%)	12 (17.4%)	0.352
6–30 min	173 (40.9%)	148 (42.3%)	24 (34.8%)	
31–60 min	71 (16.8%)	57 (16.3%)	12 (17.4%)	
Over 1 h	95 (22.5%)	74 (21.1%)	21 (30.4%)	
Heaviness of Smoking Index (missing *n* = 74)				
Low	112 (30.9%)	88 (29%)	23 (39.7%)	0.216
Medium	220 (60.6%)	187 (61.7%)	32(55.2%)	
High	31 (8.5%)	28 (9.2%)	3 (5.2%)	
Number of cigarettes/day during COVID-19 restrictions Mean (SD) (missing *n* = 8)	18 (12.1)	18.48 (12.4)	15.44 (10)	0.095
Change in smoking behaviour during COVID-19 restrictions (missing *n* = 8)				
Smoke more	190 (44.3%)	165 (46.5%)	25 (35.7%)	<0.001
No change	148 (34.5%)	135 (38%)	10 (14.3%)	
Smoke less	91 (21.2%)	55 (15.5%)	35 (50%)	
Motivation to quit prior to COVID-19 restrictions * Mean (SD) (missing *n* = 3)	5.59 (2.9)	5.27 (2.8)	7.16 (2.7)	<0.001
Self-efficacy to quit prior to COVID-19 restrictions * Mean (SD) (missing *n* = 5)	4.7 (2.9)	4.5 (2.86)	5.78 (3)	0.001
Change in motivation to quit during COVID-19 restrictions (missing *n* = 4)				
Increased considerably	86 (19.9%)	54 (15%)	31 (44.3%)	<0.001
Increased slightly	108 (24.9%)	87(24.1%)	20 (28.6%)	
No change	179 (41.3%)	169 (46.8%)	10 (14.3%)	
Decreased considerably	43 (9.9%)	36 (10%)	7 (10%)	
Decreased slightly	17 (3.9%)	15 (4.2%)	2 (2.9%)	
Change in self-efficacy to quit during COVID-19 restrictions (missing *n* = 5)				
Increased considerably	43 (10%)	25 (6.9%)	18 (25.7%)	<0.001
Increased slightly	84 (19.4%)	62 (17.2%)	21 (30%)	
No change	225 (52.1%)	208 (57.6%)	17 (24.3%)	
Decreased considerably	51 (11.8%)	43 (11.9%)	8 (11.4%)	
Decreased slightly	29 (6.7%)	23 (6.4%)	6 (8.6%)	
Changes in frequency of urges to smoke during COVID-19 restrictions (missing *n* = 6)				
Increased considerably	128 (29.7%)	106 (29.4%)	22 (31.4%)	0.009
Increased slightly	108 (25.1%)	97 (26.9%)	11 (15.7%)	
No change	128 (29.7%)	111 (30.7%)	17 (24.3%)	
Decreased considerably	50 (11.6%)	36 (10%)	14 (20%)	
Decreased slightly	17 (3.9%)	11 (3%)	6 (8.6%)	
Changes in strength of urges to smoke during COVID-19 restrictions (missing *n* = 5)				
Increased considerably	110 (25.5%)	90 (24.9%)	20 (28.6%)	<0.001
Increased slightly	120 (27.8%)	108 (29.8%)	12 (17.1%)	
No change	152 (35.2%)	133 (36.7%)	19 (27.1%)	
Decreased considerably	36 (8.3%)	23 (6.4%)	13 (18.6%)	
Decreased slightly	14 (3.2%)	8 (2.2%)	6 (8.6%)	

* on a scale of 1–10.

**Table 4 ijerph-18-01931-t004:** Factors associated with quitting during the COVID-19 restrictions.

Variable	Quit during COVID-19 Restrictions *n* (%)	Crude		Adjusted *	
		Odds Ratio (95% CI)	*p*-Value	Odds Ratio (95% CI)	*p*-Value
Education level					
12 years or less	26 (7.5%)	Ref ^		Ref	
Bachelor’s degree or higher	20 (15.2%)	2.19 (1.1, 4.0)	0.013	1.97 (1.0, 3.8)	0.048
Another smoker living at home					
Yes	13 (6%)	Ref		Ref	
No	33 (12.7%)	2.28 (1.1, 4.4)	0.016	2.18 (1.0, 4.4)	0.032
Underlying chronic illness					
No	29 (7.8%)	Ref		Ref	
Yes	17 (15.5%)	2.15 (1.1, 4.0)	0.019	2.32 (1.1, 4.6)	0.017
Perception of smokers’ risk for severe Sars-CoV-2 infection					
Same or lower risk	4 (3.9%)	Ref		Ref	
Higher risk	42 (11.1%)	3.1 (1.0. 8.8)	0.035	2.78 (0.9, 8.0)	0.06

* adjusted for all other variables, including age and sex; ^ ref- reference group.

## Data Availability

Data are available upon reasonable request from the corresponding author.

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
