# Peer review of "Changes in Smoking Behaviour and Home-Smoking Rules during the Initial COVID-19 Lockdown Period in Israel"

_ijerph, 2021, doi:10.3390/ijerph18041931_

Round 1

Reviewer 1 Report

This research investigates the change in smoking behavior during the COVID period. This is an interesting topic but substantial improvements may be required at this current form. 

1-The study enrolled random participants from social media. Can the authors elaborate on the socio-demographic characteristics of the study sample active on Facebook, Instagram, Twitter, and WhatsApp, and if the sample is representative?

2-The authors should refer to the STROBE statement in reporting their study design. The authors may avoid using the wording in the table that is not defined in the main text, for example, "Underlying chronic illness". 

3-The title is "Changes in smoking behavior and home-smoking rules during the initial COVID19 lockdown period in Israel", however, the authors failed to identify the smoking behavior before the pandemic and the "change" of smoking behavior could be hardly inferred from the questions "If 101 you used to smoke and quit, when did you quit? (since the start of the COVID19 period; 102 between 3-12 months ago - before the start of the COVID19 period but less than a year 103 ago; more than a year ago)." If the respondents quit during the covid time, that doesn't mean the covid has an impact on the respondents' behavior. There is a high possibility of reverse causality. 

Author Response

Reviewer #1

Reviewer #1

  1. The study enrolled random participants from social media. Can the authors elaborate on the socio-demographic characteristics of the study sample active on Facebook, Instagram, Twitter, and WhatsApp, and if the sample is representative?

Response:

We have added information in the Discussion regarding the proportion of Israeli adults who use these social media outlets (lines 277-279). We also state that the sample is a convenience sample and therefore unlikely to be representative of all smokers and ex-smokers in Israel (e.g., members of the ultra-orthodox Jewish community who generally do not engage in social media) (lines 279-284).:

The study sample was a convenience sample recruited through social media. In Israel, 85% of all adults’ report using Facebook, 52% report using Instagram and 15% Twitter. The sample therefore is likely to not be fully representative of all smokers and ex-smokers in Israel. More specifically, the sample was primarily restricted to Jewish participants, and females were over-represented (in comparison to their proportion among Israeli smokers). The ultraorthodox Jewish population in Israel (11% of the population), who suffered several outbreaks of SARS-CoV-2, and generally does not engage in social media, is not represented in this survey.

2. The authors should refer to the STROBE statement in reporting their study design.

Response:

We now mention in the abstract (line 8) and in the Methods section under Design (Line 59) that this was a cross-sectional study (according to the STROBE statement).

3. The authors may avoid using the wording in the table that is not defined in the main text for example "Underlying chronic illness"

Response:

We have revised the wording to maintain consistency between the main text and the table.

4. The title is "Changes in smoking behavior and home-smoking rules during the initial COVID19 lockdown period in Israel", however, the authors failed to identify the smoking behavior before the pandemic and the "change" of smoking behavior could be hardly inferred from the questions "If you used to smoke and quit, when did you quit? (since the start of the COVID19 period; between 3-12 months ago - before the start of the COVID19 period but less than a year ago; more than a year ago)." If the respondents quit during the covid time, that doesn't mean the covid has an impact on the respondents' behavior. There is a high possibility of reverse causality.

Response:

We thank the reviewer for this insightful comment. The assessment of changes in smoking behaviour was based on several questions - one regarding quitting during the initial COVID-19 period and another regarding changes in smoking frequency among current smokers only [“Since the beginning of the COVID-19 period in Israel: a) I smoke more than usual; b) I smoke the same amount; c) I smoke less than usual”]. This is stated in the Methods section. We have also added the full (translated) survey as a supplemental file.

The questions assessing quitting and quit attempts were indeed not specific to quitting due to COVID-19 and people may have quit for other reasons that were not explored. To clarify this, we have added the following text to the Discussion (lines 252-255):

“Survey questions regarding quit attempts were not specific to quitting due to COVID-19 and participants may have quit, or attempted to quit, for other reasons. Nonetheless, the baseline level of quit attempts per year among smokers is approximately 20%, suggesting that at least some of the increase was due to COVID-19.”

Reviewer 2 Report

This is a well-written article in which the authors studied two aspects of smoking behaviour (changes in smoking behaviour and home-smoking rules) at the beginning of the COVID-19 lockdown period in Israel. The study was carried out through an on-line survey from March 6 to April 28. It would be important to include the questionnaire(s) as supplementary file(s) to help readers understand how the questions were structured, as well as the type of questions included in the survey.

In my opinion, the study was correctly designed, sample size was adequate, statistical analyses seem correct, and data shown in Table 4 indicate that the factors associated with quitting smoking during COVID-19 restrictions were higher education, not living with a smoker in their home, and having a chronic disease.

Interestingly, the survey reported both an increase in smoking behaviour (probably related to stress levels) and an increase in motivation to quit during COVID-19 restriction period. It might be possible that factors such as marital status, employment status, and change in employment status (for example a divorce, reduced income or loss of employment) could be a source of stress, however those two groups (current smokers and smokers who quit during the restriction period) showed similar results regarding these factors, and it is intriguing which (other) factors could increase stress levels only in those smokers who increased smoking behaviour. Probably this issue deserves further discussion.

The authors mention and discuss the limitations and strengths of the study, and the conclusion is supported with the data presented.

Line 109: “Smoking patters” is repeated twice.

Author Response

Reviewer #2

  1. This is a well-written article in which the authors studied two aspects of

smoking behaviour (changes in smoking behaviour and home-smoking rules) at the beginning of the COVID-19 lockdown period in Israel. The study was carried out through an on-line survey from March 6 to April 28. It would be important to include the questionnaire(s) as supplementary file(s) to help readers understand how the questions were structured, as well as the type of questions included in the survey.

Response:

A supplementary file containing the survey questionnaire (translated from Hebrew to English) has been appended as suggested.

2. Interestingly, the survey reported both an increase in smoking behaviour (probably related to stress levels) and an increase in motivation to quit during COVID-19 restriction period. It might be possible that factors such as marital status, employment status, and change in employment status (for example a divorce, reduced income or loss of employment) could be a source of stress, however those two groups (current smokers and smokers who quit during the restriction period) showed similar results regarding these factors, and it is intriguing which (other) factors could increase stress levels only in those smokers who increased smoking behaviour. Probably this issue deserves further discussion.

Response:

We thank the reviewer for this insightful comment. The current study focused on factors associated with quitting during the initial COVID19 restrictions period, and did not explore in-depth the factors associated with increased smoking frequency. Regarding the association between stress levels and quitting, it was surprising to find that pre-COVID19 stress levels and perceived change in stress levels during the COVID19 restrictions period, were similar among those who quit and those who continued to smoke. We have added this to the Discussion (lines 246-251):

“Having a higher educational degree, not living with another smoker in the home, and having an underlying chronic illness, were all associated with quitting during the initial COVID19 restrictions period. Surprisingly, pre- COVID19 stress levels, and perceived changes in stress levels during the COVID19 restrictions period, were similar among current smokers and those who quit.”

Other factors that may have impacted quitting and were not explored in our research were certain smoking characteristics such as number of cigarettes per day, level of nicotine dependence, baseline level of perceived self-efficacy and motivation to quit. Unfortunately, as stated in the Discussion section (lines 285-288), the survey was designed in a way that collected smoking characteristics only among current smokers, preventing us from exploring these associations. However, data presented in Table 3 show that there were significant differences in smoking characteristics between current smokers who attempted to quit and those who did not. This supports the hypothesis that these factors play a significant role in driving quitting behaviour, more than stress levels themselves. We have added this to the Discussion (lines 287-291):

“Certain smoking characteristics such as a lower nicotine dependence level, higher perceived self-efficacy to quit, and higher baseline motivation level might be associated with a greater likelihood of successful quitting during the initial COVID19 restrictions period. Some evidence for this can be seen from findings presented in Table 3 comparing current smokers who attempted to quit and those that did not attempt to quit.”

3. Line 109: “Smoking patters” is repeated twice.

Response:

            This has been corrected.

Reviewer 3 Report

It is an interesting survay about the relationship Covid-19 and smoke.

I suggest some changes:

in materials and methods section please point out that it it a retrospective study and its main endpoints.Concerning smoking status I suggest to include further 

parameters if possible, such as cigarette number a day average and SD, pack-years level that is indicative of smoke exposure. Which test was used to evaluate the motivational value? please add a statement about.

Are there data avaluable concerning the percentage of relapse for quitters during quarantine?Please change analysis with statistical analysis.

In the results section please include in the paragraph the data regarding the employment status change reported in table 1 since the difference was significant.

I eventually suggest to add some references to widen the discussion:

-J Med Virol. 2020 Oct 7:10.1002/jmv.26585. doi: 10.1002/jmv.26585. about the influence of smoking and COPD on Covid-19 survival

-Acta Biomed. 2020 Aug 27;91(3):e2020062. doi:

10.23750/abm.v91i3.10373. about the role of smoke in Covid-19

-Tuberk Toraks. 2020 Dec;68(4):371-378. doi: 10.5578/tt.70352.smoking status in hospetalized patients

- J Thorac Dis. 2020 Dec;12(12):7429-7441. doi: 10.21037/jtd-20-1743.about risk factors

Author Response

Reviewer #3

  1. In materials and methods section please point out that it is a retrospective study and its main endpoints.

Response:

We have added the information that this study was a cross-sectional study (not retrospective) (line 59). In addition, we have added the main study outcomes to the Methods section (lines 65-67):

“Main study outcomes: 1) quitting during the initial COVID-19 restrictions period in Israel; 2) changes in home smoking rules during the initial COVID-19 restrictions period in Israel; and 3) changes in smoking behavior among current smokers.”

2. Concerning smoking status I suggest to include further parameters if possible, such as cigarette number a day average and SD, pack-years level that is indicative of smoke exposure.

Response:

The Methods section details all the smoking-related variables that were assessed, including average number of cigarettes per day (mean and SD, prior to and during the initial COVID-19 restrictions period) (lines 115-119). The survey did not include a question on the number of years the respondent has been smoking so we are unable to assess pack-years. We have now included this as a limitation in our discussion (lines 294-295):

“In addition, the survey did not include a question on the number of years the respondent has been smoking so we are unable to assess pack-years.”

3. Which test was used to evaluate the motivational value? please add a statement about.

Response:

Motivation to quit smoking was assessed with the question “Before the beginning of the COVID-19 pandemic period in Israel, how do you estimate your level of motivation to quit smoking was, from 1-10 (1-did not want to quit smoking at all; 10-very much wanted to quit smoking)”. This information is presented in Table 3 as mean and SD. The difference in the mean level of motivation to quit between current smokers who attempted to quit and those who did not (during the COVID-19 restrictions period) was assessed using t-test. We have added the survey questionnaire in a supplemental file, and the information regarding the statistical tests that were used is described in the Methods section (lines 143-148)

4. Are there data available concerning the percentage of relapse for quitters during quarantine?

Response:

Unfortunately, the survey did not assess relapse among former smokers. We agree this is important information that should be included in any future studies. We have added this to the Discussion (lines 292-294):

“Data regarding the proportion of former smokers who relapsed during the initial COVID-19 restrictions period were not collected. It is therefore not possible to explore which factors might impact relapse.”

5. Please change analysis with statistical analysis.

Response:

This has been changed.

6. In the results section please include in the paragraph the data regarding the employment status change reported in table 1 since the difference was significant.

Response:

We have added this information to the Results as suggested (lines 173-176):

“Compared to current smokers and to those who quit during the initial COVID-19 restrictions period, ex-smokers (who quit prior to the COVID-19 restrictions) tended to be older, male, living with a partner, with their employment status unchanged during the initial COVID-19 re-strictions period, (Table 1).”

7. I eventually suggest to add some references to widen the discussion:

J Med Virol. 2020 Oct 7:10.1002/jmv.26585. doi: 10.1002/jmv.26585. about the influence of smoking and COPD on Covid-19 survival

Acta Biomed. 2020 Aug 27;91(3):e2020062. doi: 10.23750/abm.v91i3.10373. about the role of smoke in Covid-19

Tuberk Toraks. 2020 Dec;68(4):371-378. doi: 10.5578/tt.70352.smoking status in hospetalized patients

J Thorac Dis. 2020 Dec;12(12):7429-7441. doi:  10.21037/jtd-20-1743.about

risk factors

Response:

We thank the reviewer for these references. We have elaborated on this point in the Discussion, mentioning the effect that the uncertainty regarding the role smoking and nicotine might play in the pathogenesis of COVID-19, and referred to these articles (lines 267-276):

“The uncertainty regarding the role of smoking and nicotine on the risk for infection and COVID-19 severity might also impact smoking behaviors. Several meta-analyses have suggested that former smokers are at increased risk for a severe disease, while other studies have shown that the proportion of smokers among infected patients was lower than expected. This has raised the hypothesis that nicotine might play an immune-modulator protective role in the pathogenesis of the disease. Claims based on this hypothesis were made on popular media outlets which might impact smoking behavior. Future studies assessing changes in smoking behavior need to also assess exposure to these claims and the impact of this exposure on the changes in smoking behavior.”  

Round 2

Reviewer 1 Report

The current format is acceptable based on the revision the authors submitted. The professional editing services is highly recommended before publication.